# Effect of Guanylate Cyclase-22-like on Ovarian Development of *Orius nagaii* (Hemiptera: Anthocoridae)

**DOI:** 10.3390/insects15020110

**Published:** 2024-02-04

**Authors:** Huiling Du, Ruijuan Wang, Xiaoyan Dai, Zhenjuan Yin, Yan Liu, Long Su, Hao Chen, Shan Zhao, Li Zheng, Xiaolin Dong, Yifan Zhai

**Affiliations:** 1MARA Key Laboratory of Sustainable Crop Production in the Middle Reaches of the Yangtze River (Co-Construction by Ministry and Province), College of Agriculture, Yangtze University, Jingzhou 434025, China; 2Institute of Plant Protection, Shandong Academy of Agricultural Sciences, 23788 Gongye North Road, Jinan 250100, China; 3Key Laboratory of Natural Enemies Insects, Ministry of Agriculture and Rural Affairs, Jinan 250100, China

**Keywords:** *Orius nagaii*, receptor-type guanylate cyclase-22-like (GCY-22), vitellogenin (Vg), RNA interference, reproduction

## Abstract

**Simple Summary:**

In this study, the effect of the targeted silencing of *OnGC*Y on the reproduction of the pirate bug *Orius nagaii* was evaluated. *OnGCY* silencing significantly inhibited ovarian development and egg production and significantly reduced the expression of the gene encoding the vitellogenin (*Vg*) gene. These data provide insights into the function of *OnGCY* in insect reproduction, which could guide research for the mass rearing of natural enemies for the biological control of insect pests.

**Abstract:**

This study identified and characterized the gene encoding recep tor-type guanylate cyclase-22-like (GCY-22; *OnGCY*) from the pirate bug *Orius nagaii*, an important biological control agent. The full-length cDNA of the *GCY* of *O. nagaii* was obtained by rapid amplification of cDNA ends (RACE); it had a total length of 4888 base pairs (bp), of which the open reading frame (ORF) was 3750 bp, encoding a polypeptide of 1249 amino acid residues. The physicochemical properties of *OnGCY* were predicted and analyzed by using relevant ExPASy software, revealing a molecular formula of C6502H10122N1698O1869S57, molecular weight of ~143,811.57 kDa, isoelectric point of 6.55, and fat index of 90.04. The resulting protein was also shown to have a signal peptide, two transmembrane regions, and a conserved tyrosine kinase (tyrkc). Silencing *OnGCY* by RNA interference significantly inhibited ovarian development and decreased fertility in female *O. nagaii* in the treated versus the control group. Additionally, *OnGCY* silencing significantly decreased the expression levels of other *GCY* and *Vg* genes. Thus, these results clarify the structure and biological function of *OnGCY*, which has an important role in insect fecundity. The results also provide a reference for agricultural pest control and future large-scale breeding of biological control agents.

## 1. Introduction

*Orius* spp. (Hemiptera: Anthopteridae) originated in Asia but are now distributed worldwide. As an important euryphagous predator, they are widely used to control small pests, such as aphids, thrips, mites, and some lepidopterans, on crops in fields and greenhouses [1,2,3,4,5]. In Asia, it is an important biological control agent for agricultural and forestry insect pests [6]. It can prevent and control aphids and whiteflies on fruits and vegetables and suck dry prey through pricking mouthparts. Reduce the pollution of chemical pesticides in the field, achieve the effect of green prevention and control, ensure the quality of agricultural products, and have great potential for field application [7]. Indoor breeding of *Orius nagaii* showed that it has strong adaptability and high spawning capacity, making it relatively straightforward to breed this species in large quantities, which has significant potential for commercial breeding [8].

Receptor-type guanylate cyclase (GYC) is a guanylate cyclase (GC) receptor with ligand-binding sites in its extracellular domain and an intracellular catalytic domain that catalyzes the production of cyclic guanosine monophosphate (cGMP) from guanosine triphosphate (GTP). cGMP activates cGMP-dependent protein kinase G (PKG), resulting in the phosphorylation of serine or threonine residues of specific proteins, causing cellular responses [9]. The second messenger of this pathway is cGMP. At least two forms of GC are present inside the cell: membrane-bound GC (mGC) and soluble GC (sGC). mGC is a transmembrane protein with an extracellular N terminus, which is the kinase-binding region. The remainder of the protein is intracellular and contains a protein-like kinase region and a catalytic region at the C terminus. mGC forms part of the main enzyme-linked receptor signaling pathway. Different receptors can bind to different first messenger hormones, which can be divided into GC-A, GC-B, and GC-C according to how they bind to ligands [10,11,12,13,14]. GC-A binds cardiac natriuretic peptides [15,16], GC-B binds brain natriuretic peptides [17], and GC-C binds bacterial heat-stable enterotoxin [12]. All membrane-bound GCs contain a protein kinase-like domain with high homology to protein kinase located between the plasma membrane and the catalytic region of GC [18]. mGC is similar to growth factor receptors that show tyrosine protein kinase activity, although the protein kinase activity of mGC remains to be confirmed [15]. By contrast, cytoplasmic sGC, a cytosolic protein tightly bound to heme and activated by nitric oxide (NO), is a GC that differs significantly from mGC [19].

The role of *GCY* in the reproductive development of insects is not yet fully understood. However, deletion of *GCY-35* or *GCY-36* from *Caenorhabditis elegans* resulted in small body size, developmental delay, and exploration deficits exhibited by multiple Bardet-Biedl syndrome mutants [20]. The activity of *GCY-33*, an atypical sGC oxygen sensor, in neurons is essential for extending the lifespan of *C. elegans* [21]. *GCY* is not only involved in growth and development but also has applications in neurology. Hallem et al. reported that CO2 can specifically activate BAG neurons in *C. elegans* and that the CO2-sensitive function of BAG neurons requires TAX-2/TAX-4 cyclic nucleotide-gated ion channels and *GCY-9*. They proposed that the activation of *GCY* is an evolutionary conserved mechanism for the detection of environmental CO2 in animals [22]. Similarly, mutant *C. elegans* lacking *GCY-28* did not approach the odor perceived by primary chemical sensory neurons in the pharynx and cannot associate benzaldehyde with starvation. The novel role of *GCY-28* in associative memory retrieval could have broad implications for neural mechanisms of learning and memory [23]. Furthermore, salt-competent *GCY-22* in *C. elegans* accumulates at high concentrations in subcompartments in the distal region of cilia. The ciliary tip targeting of *GCY-22* is dynamic and requires an intraflagellar transport (IFT) system to regulate protein transport in and out of cilia, disruption of which leads to mislocalization of GCY-22 proteins and defects in the formation and maintenance of the ciliated apical septum [24].

In this study, the gene encoding *OnGCY* was cloned and identified, and its effects on *O. nagaii* ovarian development and oviposition were studied. In addition, real-time quantitative (q)PCR was used to detect the expression of *OnGCY* in different developmental stages and tissues of *O. nagaii*. The results of this study will be useful to increase understanding of the effect of *OnGCY* on the reproductive development of *O. nagaii*. In addition, they could help support the mass rearing of *O. nagaii* for use as an effective biological control agent.

## 2. Materials and Methods

### 2.1. Insects 

*O. nagaii* were obtained from the natural enemy greenhouse of the Shandong Academy of Agricultural Sciences (Jinan, China). They were reared on *Phaseolus vulgaris* L. and *Sitotroga cerealella* eggs in the laboratory under specific environmental conditions (temperature 26 °C; relative humidity 60–70%; photoperiod 16 h light: 8 h dark; 12 W energy-saving lamp). The replacement prey was fresh *S. cerealella* eggs irradiated with ultraviolet light for 30 min, and kidney beans were used as a water source and yield medium.

### 2.2. Sample Preparation

Samples of adult *O. nagaii* were collected to determine the expression levels of *OnGCY* in different developmental stages and tissues. In phosphate-buffered saline (PBS; PH = 7.4), the head (n = 30), thorax (n = 30), abdomen (n = 30), intestine (n = 30), and ovary (n = 30) of female adults just emerging for 24 h were dissected with cell tweezers. The dissected tissues were washed with clean PBS. Next, samples of *O. nagaii* at different developmental stages were collected using homemade fluke tubes as follows: egg (n = 40), 1st instar naiad (n = 40), 2nd instar naiad (n = 40); 3rd instar naiad (n = 40), 4th instar naiad (n = 30), 5th instar naiad (n = 30), female adults (n = 20), and male adults (n = 20). All samples were collected and placed in a 1.5 mL RNAse-free microcentrifuge tube. Then, they were immediately frozen in liquid nitrogen and stored in a −80 °C refrigerator for RNA extraction. Total RNA was extracted, reverse transcription was performed, cDNA was used as a template, Actin was used as internal control, and real-time PCR was performed. For each developmental stage and each tissue, three biological replicates are performed.

### 2.3. Total RNA Extraction and cDNA Synthesis

Total RNA was extracted from samples collected from different developmental stages and tissues of *O. nagaii* using the FsatPure^®^Cell/Tissue Total RNA Isolation Kit V2 (Vazyme, Nanjing, China) according to the manufacturer’s instructions. The kit can be operated under normal temperature conditions. The sample was placed in a 1.5 mL RNase-free centrifuge tube with 500 µL Buffer RL and ground with an electric grinder. The intermediate link can be operated according to the manual. The adsorption column was transferred to a new 1.5 mL RNase-free centrifuge tube, and 40 µL of RNase-free ddH2O was added to the center of the adsorption column, left for 1 min, centrifuged at 12,000 rpm for 1 s, and RNA was eluted. The extracted RNA was stored in a −80 °C refrigerator. 

The concentration of total RNA was determined using a NanoDrop 2000 spectrophotometer (Thermo Fisher Scientific, Waltham, MA, USA). The RNA integrity was evaluated by 1% agarose gel electrophoresis. The cDNA synthesis using a PrimeScript™ RT reagent Kit with gDNA Eraser (Perfect Real Time) (TaKaRa, Dalian, China). Briefly, 1 µg total RNA in a 20 µL reaction mixture, which was stored at −20 °C until use. Each kit was used according to the manufacturer’s instructions. 

### 2.4. OnGCY Gene Cloning

A pair of primers, On-GCY-F/On-GCY-R, was designed based on *OnGCY* (Table 1). The extracted total RNA was subjected to RT-PCR using Phanta Max Fidelity DNA Polymerase (Vazyme, Nanjing, China) according to the manufacturer’s instructions. The amplification product was sequenced to obtain the core sequence. The PCR conditions were as follows: predenaturation at 95 °C for 30 s/3 min, then 30 cycles of denaturation at 95 °C for 15 s, annealing at 60–65 °C for 15 s, extension at 72 °C for 1 min, and finally extension at 72 °C for 5 min. 

Rapid amplification of cDNA ends (RACE) primers were then designed, and 5′RACE or 3′RACE was performed based on the core sequence. 5′RACE was synthesized according to Roche’s 5′/3′RACE 2nd Generation kit (Roche, Shanghai, China), using total RNA, GCY-5′RACE-rt-r, and Control Primer neo1/rev (Vial 10) for reverse transcription with incubation at 55 °C for 60 min, and then at 85 °C for 5 min. The Plasmid Mini Kit I (200) (Omega Biotek, Shanghai, China) was used to purify the cDNA, which was then used directly for poly(A) tailing of the first strand of cDNA by terminal transferase.

The first PCR was performed using GCY-5′RACE-r1 and Oligo dT-Anchor Primer (Vial 8) as follows: predenaturation at 94 °C for 2 min, followed by 30 cycles of denaturation at 94 °C for 15 s, annealing at 60 °C for 30 s, extension at 72 °C for 40 s, and extension at 72 °C for 7 min. Using GCY-5′RACE-r2-1 or GCY-5′RACE-r2-2 primers and PCR Anchor Primer (Vial 9), nested PCR was performed under the same conditions using the first round of PCR products as a template, and the annealing temperature was increased to 60–65 °C. The cDNA fragment was obtained using Phanta Max Fidelity DNA polymerase (Vazyme, Nanjing, China).

The synthesis of 3′RACE cDNA was performed using a 3′full Race Core Kit (TaKaRa, Dalian, China) according to the manufacturer’s instructions, using GCY-3′RACE-f1 primers and a 3′RACE Outer Primer for the first PCR with the following steps: predenaturation at 94 °C for 3 min, then 30 cycles of denaturing at 94 °C for 30 s, annealing at 60 °C for 30 s, extension at 72 °C for 1 min, and finally an extension at 72 °C for 10 min. Nested PCR was then performed using GCY-3′RACE-f2-1 or GCY-3′RACE-f2-2 primers and a 3′RACE Control Inner Primer, using the first round of PCR products as a template. The resulting fragment was obtained using Phanta Max Fidelity DNA polymerase (Vazyme, Nanjing, China). The RACE product was electrophoresed on 1% agarose gel at 180 V for 20 min; then, the gel was cut and purified. The recovered products are sequenced directly. The 5′RACE, core sequence, and 3′RACE were spliced together to obtain the full-length sequence of OnGCY cDNA. The RACE trial was completed by Shanghai Zhishuo Biotechnology Co., Ltd. (Shanghai, China) (http://www.pszhishuo.com/, accessed on 8 September 2022).

### 2.5. OnGCY Genetic Analysis

DANMAN V6.0 (Lynnon Biosoft, San Ramon, CA, USA) software was used to edit sequencing results and sequence splicing and for multiplex comparison of sequences. The translated amino acid sequences were subjected to homology comparison and conserved region analysis using BLAST (https://blast.ncbi.nlm.nih.gov/Blast.cgi, accessed on 4 January 2023). The Sigcleave program on EMBOSS Explorer (http://emboss.toulouse.inra.fr/cgi-bin/emboss, accessed on 4 January 2023) was used to predict signal peptides. Transmembrane regions were predicted using the NovoPro online tool (www.novopro.cn/tools/tmhmm.html, accessed on 6 January 2023), whereas protein transmembrane region prediction and functional regions were predicted using the Simple Modular Architecture Research Tool (SMART; http://smart.embl.de/, accessed on 6 January 2023). Molecular weight was calculated using the Compute pI/Mw sequence of ExPASy (http://web.expasy.org/compute-pi, accessed on 10 January 2023), and the GCY protein domain was analyzed using the Conserved Domain Database (www.ncbi.nlm.nih.gov/Structure/cdd/wrpsb.cgi, accessed on 11 January 2023).

### 2.6. OnGCY Expression in Different Developmental Stages and Tissues

An online website (www.primer3plus.com/index.html, accessed on 25 January 2023) was used to design qPCR primers (Table 1) based on the previously obtained *OnGCY* sequence. qPCR analysis was performed on a Quantstudio 6 Flex 7500 Fast Real-time PCR system (Thermo Fisher Scientific) using 2 × SYBR Green qPCR Mix (With ROX) (SparkJade, Jinan, China) to detect the expression levels of *OnGCY* in the different developmental stages and tissues of *O. nagaii*. The qPCR reaction mixture comprised of 10 µL 2 × SYBR qPCR Mix, 0.4 µL ROX Reference Dye II, 0.4 µL each of the forward and reverse primers, and 2 µL *O. nagaii* cDNA template, made up of a final volume of 20 µL using ddH_2_O. The cycling conditions were as follows: predenaturation at 95 °C for 20 s, 40 cycles of denaturation at 90 °C for 3 s, and annealing at 60 °C for 30 s. Reactions were performed in quadruplicate for each sample, and gene expression levels were normalized against *O. nagaii* actin according to the CT method (2^−∆∆Ct^) [25]. Three biological replicates and four technical replicates were performed for each developmental stage and tissue.

### 2.7. Double-Stranded RNA Synthesis

Based on the previously obtained *OnGCY* sequence, gene-specific primers containing the T7 polymerase promoter sequence were designed using Primer3Plus (www.primer3plus.com/index.html, accessed on 18 January 2023) (Table 1). For double-stranded RNA synthesis, a T7 RiboMAX^TM^ Express RNAi System (Promega) was used. With the guide of the kit, the previously synthesized first-strand cDNA was used as a DNA template. The primers in Table 1 were used for PCR amplification containing the T7 promoter sequence. The concentration of the synthesized dsRNA was calculated by a NanoDrop 2000 spectrophotometer, and its integrity was estimated by 1% agarose gel electrophoresis. The green fluorescent protein (GFP) gene was used to synthesize GFP dsRNA and was generated as a positive control.

### 2.8. RNAi 

RNAi was used to investigate the function of *OnGCY* in *O. nagaii* reproduction (Table 1). Newly emerged (<24-h old) female *O. nagaii* adults were put in an empty culture plate that was then placed on ice to make them faint. Place the *O. nagaii* in the filter paper so that its abdomen is up near the direction of the glass needle. A NANOJECT II microinjector (Gairdner, Wuhai, China) was then used to inject 50 nL *dsGCY* (more than 5000 µg/µL) into the middle part of the midfoot coxa and the postfoot coxa of the female adult. Do not pierce the body.

An equal volume of *dsGFP* was injected into the negative control group in the same position. The insects were then placed in plastic culture plates containing fresh kidney bean nuggets and *Sitotroga cerealella* eggs and kept in an artificial climate chamber (temperature 26 °C; relative humidity 60–70%; photoperiod 16 h light: 8 h dark) for 1–5 days. The efficiency of RNAi was detected by qPCR, and the expression level of the GCY gene in females was determined 1–5 days after injection of dsRNA. For each treatment group with RNA interference for 1–5 d, three biological repetitions, and four technical repetitions were carried out, and ten insects were used to extract RNA each time. And dsRNA injection for 72 h to detect the expression of the Vg gene. The treatment was performed in three biological replicates, four technical replicates, and ten insects were used to extract RNA per biological replicate.

### 2.9. Bioassay

To assess the effect of RNAi on the fertility of *O. nagaii*, males and females of newly emerged *O. nagaii* were mated in pairs for 24 h, after which the females were injected with either *dsGCY* or *dsGFP* (as controls) as described in the previous section. The females were then placed in plastic culture plates and supplied with kidney bean nuggets and *Sitotroga cerealella* eggs. Egg production was counted daily, and kidney bean nuggets were replaced each day over a period of 14 days. Three biological replicates were performed for each treatment, with 20 adult females for each replicate (Valid data is not less than 50). Ovaries were dissected 24, 48, or 72 h after injection and washed with PBS before using an ultra-mirror deep 3D microscope (VHX-6000, KEYENCE, Shanghai, China) to observe, compare, and photograph ovarian development. Five biological replicates were applied per treatment.

### 2.10. Statistical Analysis

The 2^−∆∆Ct^ method [25] was used to calculate the relative expression of *GCY* and *Vg* genes. All experimental data were analyzed using PASW Statistics 18 software (SPSS Inc., Chicago, IL, USA). The Student’s *t*-test for independent samples was used to analyze the significance of the efficiency of RNAi silencing. Mann-Whitney U was used to compare the total egg size and the pre-oviposition period. Significant differences between the treatments were assessed using a one-way analysis of variance, followed by the Tukey test for multiple comparisons. Differences were significant at *p* < 0.05.

## 3. Results

### 3.1. Sequence Identification and Characteristics of OnGCY

The 5′ and 3′ terminal sequences of *OnGCY* were obtained by cloning of 5′ and 3′ FULL RACE, and the known partial sequences were analyzed in combination with transcriptome to obtain the full-length ORF and the 5′ and 3′ noncoding sequences of *OnGCY*. According to the verification results, the cDNA of *OnGCY* was 4888 base pairs (bp), the final full-length ORF was 3747 bp, and encoded a hypothetical protein sequence of 1249 amino acids. The ProtParam program on ExPASy (https://www.expasy.org/resources/protparam, accessed on 15 January 2023) was used to obtain the physicochemical properties of the protein. The molecular formula of the *OnGCY* was C6502H10122N1698O1869S57, with a molecular weight of ~143.811 kDa and a theoretical isoelectric point of 6.55. The estimated half-life of *OnGCY* was 30 h, with an instability index (stable protein < 40) of 42.89 and an aliphatic index of 90.04. The average hydropathicity of *OnGCY* was −0.239, with a total number of negatively charged residues (Asp + Glu) of 149 and a total number of positively charged residues (Arg + Lys) of 144. The SMART tool showed that *OnGCY* had a low complexity region (Low), a transmembrane region ™, and a conserved tyrosine kinase domain (tyrosine kinase, catalytic domain; Tyrkc), followed by another area of low complexity (Figure 1).

### 3.2. Prediction of OnGCY Protein Domains

The degree of similarity between the GCY protein of *O. nagaii* and that of other species was determined using a BLAST homology search and comparison analysis. *OnGCY* showed 95% similarity with *Amphimedon queenslandica* (GenBank accession number: XP 019848885.1) and *Hydractinia symbiolongicarpus* (GenBank accession number: XP 057304139.1), respectively. Furthermore, there was 100% similarity with the coleopteran *Photinus pyralis* (GenBank accession number: XP 031356046.1). The phylogenetic tree (Figure 2) showed that the *On*GCY clustered with *A. queenslandica*, *H. symbiolongicarpus*, and *P. pyralis*, being most closely related to that of *P. pyralis*.

### 3.3. OnGCY Expression in Different Developmental Stages and Tissues

The expression pattern of *OnGCY* in the different developmental stages of *O. nagaii* was detected by qPCR. *OnGCY* was expressed at fluctuating levels in all developmental stages of *O. nagaii* and in various tissues of *O. nagaii* adults. The levels were highest in the abdomen, followed by the ovary and intestine, and lowest in the thorax and head (Figure 3A). There are differences in the expression of OnGCY among eggs, 1–5 instars, nymphs, and adults. OnGCY has the highest expression in eggs. In the nymph stage, the expression level was the highest in the 2nd instar nymph and the lowest in the 1st, 3rd, 4th, and 5th instar nymph. There was no significant difference between 1st, 3rd, 4th, and 5th instar nymphs, but there was a significant difference between 2nd instar nymphs and 1st, 3rd, 4th, and 5th instar nymphs. There was no significant difference in the expression level between mating females (m-F) and mating males (m-M) at the adult stage (Figure 3B). 

### 3.4. Effects of RNAi Silencing on OnGCY Gene Expression in Adult Female O. nagaii

The level of *OnGCY* mRNA was determined 1–4 days after injecting newly emerged females with *dsGCY*. Compared with the control group (injected with *dsGFP*), the expression of *dsGCY* was downregulated by 96.16%, 99.71%, 86.16% and 62.6299% 1–5 days after injection, respectively. The downregulation was the most obvious on days 2 and 3 (Figure 4). This suggests that *dsGCY* injection significantly suppressed the mRNA level of *OnGCY* and that this suppression lasted for at least 4 days, leading to an effective RNAi response.

### 3.5. Effect of Silencing of OnGCY on Reproduction of O. nagaii Females

To investigate the effect of *OnGCY* on the growth and development of *O. nagaii*, dsRNA was injected into male and female insects, which were then reared on kidney bean nuggets and *S. cerealella* eggs, and the preoviposition period, total number of eggs laid, and average daily number of eggs laid were recorded. Compared with the control group (*dsGFP* injection), the number of eggs laid by females in the *dsGCY* group decreased by 36.51%, with a significant difference between the two groups (Figure 5B). The preoviposition period of the *dsGCY* group was significantly longer than that of the control group, and the spawning time was delayed (Figure 5A). Compared with the control group, there was a statistically significant difference in the daily egg production of *dsGCY* females at 3–9 days post-injection, and the average daily egg production of the control group increased significantly at 3–9 days post-injection. Egg production peaked at 6 days post injection; the total egg production of *dsGCY* females was relatively low, with the highest egg production at 6–9 days post injection (Figure 5C). Compared with the control group, there was a significant difference in the average daily number of eggs laid by *dsGCY* females at 3–9 days post injection; the average daily number of eggs laid by *dsGCY* females first increased and then decreased, being highest at 5–9 days post-injection. By contrast, the oviposition peak in the control group was 6 days post-injection, and they laid a higher average daily number of eggs compared with *dsGCY* females. 

### 3.6. Effect of Silencing of OnGCY on Ovarian Development of O. nagaii Female Adults

To further understand the effect of *OnGCY* on the ovarian development of *O. nagaii*, ovaries were dissected 24 h, 48 h, and 72 h after *dsGCY* injection, and their appearance was examined under a microscope. In addition, the expression of *OnVg* 72 h after interference was also determined. The control group was injected with *dsGFP*. Compared with the control group, the expression of *OnVg* 3 days after injection was significantly downregulated (by 57.67%), indicating that the expression of *OnVg* was reduced by interfering with *OnGCY* (Figure 6A). Fewer eggs were recorded within the ovaries of females 24 h after injection with *dsGCY*. Mature eggs were observed in the control group at 48 h compared with 72 h in the treatment group. The eggs in the latter group were also significantly smaller than in the control group (Figure 6B). Substantial mature oocytes were observed in the *dsGFP* group, whereas only a few mature oocytes were observed in the *dsGCY*-treated group.

## 4. Discussion

GCY is a common enzyme that catalyzes the conversion of GMP to cGMP. The latter has an important role in regulating signaling molecules in cells and is involved in the regulation of various physiological processes, such as visual conduction, vasodilation, cardiovascular function, and nerve conduction, among others [26,27]. GCY enzymes have different molecular structures and functional characteristics in different organisms. In animals, GCY is mainly divided into two types: soluble GCY (sGCY) and membrane-bound GCY (mGCY). sGCY is widely found in a variety of tissues and cells, including the heart, lungs, and blood vessel walls, and is regulated by NO and some hormones. mGCY is mainly found in visual cells and is a key enzyme in visual conduction [28]. In addition, *GCY-22* is a salt-sensitive GCY in *C. elegans* [23].

In this study, based on the *OnGCY* transcriptome sequence, RT-PCR combined with RACE was used to clone the full-length cDNA sequence of *GCY* from *O. nagaii*, the structure of which was then determined. GCYs contain multiple sites, such as a catalytic site between amino acids 1 and 1200. Periplasmic binding protein (PBP) 2 has the same structure as type 1 exoplasm binding protein (PBP1) but a different topology. PBP2 is involved in the uptake of multiple soluble substrates, such as phosphate, sulfate, polysaccharides, lysine/arginine/ornithine, and histidine [29]. After binding to specific ligands with high affinity, PBP2 interacts with homologous membrane transport complexes comprising two intact membrane domains and two ATPase domains located on the cytoplasm. γ-Aminobutyric acid (GABA) is the main inhibitory neurotransmitter in the mammalian central nervous system, and GABAb receptors are members of the receptor superfamily that includes mGlu receptors. Coupling of the GABAb receptor to a G protein leads to a decrease in calcium, an increase in potassium membrane conductance, and inhibition of cAMP formation [30,31]. Receptor tyrosine kinase (RTK) is a membrane protein containing extracellular ligand-binding regions, transmembrane segments, and intracellular tyrosine kinase domains. It is normally activated by ligand binding, resulting in dimerization and autophosphorylation of intracellular Tyrkc domains, leading to intracellular signaling. PTKs have an important role in many cellular processes, including lymphocyte activation, epithelial growth and maintenance, metabolic control, organogenetic regulation, survival, proliferation, differentiation, migration, adhesion, motility, and morphogenesis [32].

The expression levels of *OnGCY* in different developmental stages and tissues of *O. nagaii* were detected by qPCR. *OnGCY* was expressed at fluctuating levels in all developmental stages of *O. nagaii*, with expression being highest in 2nd-instar nymphs. *OnGCY* expression was detected in all tissues, including the ovaries, suggesting that GCY is essential for *O. nagaii* reproduction. In mammals, receptor guanylyl cyclase is involved in the regulation of many physiological processes. It comprises receptor domains and catalytic domains that bind to specific ligands through the receptor moiety, converting GTP to cGMP through the enzymatic moiety [28,33]. cGMP is an important cell signaling molecule with a key role in regulating cell growth, differentiation, metabolism, and signal transduction. RGC is particularly important in the reproductive system. For example, RGC is involved in regulating sperm formation, maturation, and motility, as well as in the regulation of follicle development and ovulation [34]. In addition, RGC is also involved in the regulation of the nervous, visual, cardiovascular, and intestinal systems. However, and there might be interspecies and intertissue differences that require further investigation to provide a more detailed understanding of the role of RGC in reproductive development across species [35].

The structure of membrane-bound GCY has been reported to resemble a class of growth factor receptors with tyrosine protein kinase activity, such as the insulin transmembrane receptor (InR) [36]. After activation, protein kinase (PDK) and protease (AKT/PKB) sequentially regulate their downstream effector proteins, affecting physiological activities, such as sugar uptake, lipid synthesis, and gene expression, with important roles in the regulation of cell growth and nutrient metabolism [37]. Vg not only provides nutrients, thus affecting the quantity and quality of insect egg production, but also has a key role in the embryonic development of oviparous animals [38], as well as participating in various biological functions and complex metabolic processes, such as immune defense and lifespan regulation [39,40]. Vg uptake is essential for ovarian development. RNAi of *Vg* in insects such as *Nilaparvata lugens* [41], *Bemisia tabaci* [42], and *Panonychus citri* [43] can inhibit ovarian development and reduce egg production, while RNAi-mediated suppression of *Vg* expression in adult *Cimex lectularius* females drastically reduced egg production, and resulted in atrophied ovaries and an inflated abdomen caused by hypertrophied fat bodies [44]. In Drosophila, GCY is associated with memory and embryonic cell development [45,46,47]. In *C. elegans*, GCY is also involved in body development, but more reports have focused on neural mechanisms [20,21]. Guanylate cyclase receptors (rGCs) have been found to be involved in the reproduction of basal metazoans in the sperm flagella of coral stones [34]. However, little research has been conducted on the GCY gene in other species, focusing more on model organisms. In this study, RNAi was used to analyze the role of *OnGCY* in *O. nagaii* reproduction. The results showed that expression of *GCY* and *Vg* after RNAi significantly decreased egg production, as well as prolonged the pre-spawning period and impacted the average daily egg production. The tyrkc conserved domain of *OnGCY* is similar to that of InR, and. might be involved in internal factors affecting insect lifespan, such as insulin-like peptide, AMP-activated protein kinase, and target of rapamycin [48,49,50,51]. Therefore, gene up- and downregulation of genes and related pathways in *O. nagaii* after *GCY* RNAi require further study, bearing in mind that this might also involve interaction with other signaling pathways.

In conclusion, *GCY* is an enzyme with important biological functions and significant research value. Study of its structure and function, regulatory mechanism, and role in disease provides not only insights into mechanisms of cell signaling but also theoretical support for related basic applications

## 5. Conclusions

In summary, we identified *OnGCY* in the *O. nagaii* genome and transcriptome. Bioinformatics analysis revealed that *OnGCY* has a conserved tyrosine kinase domain. *OnGCY* silencing significantly inhibited ovarian development and fecundity of female *O. nagaii* and significantly inhibited the transcription level of *Vg*. These data could help clarify the physicochemical properties, signal peptides, transmembrane regions, protein domains, and biological functions of *OnGCY*, furthering understanding of the role of *OnGCY* in insect reproduction and providing a reference for the large-scale development of this important biological control agent.

## Figures and Tables

**Figure 1 insects-15-00110-f001:**
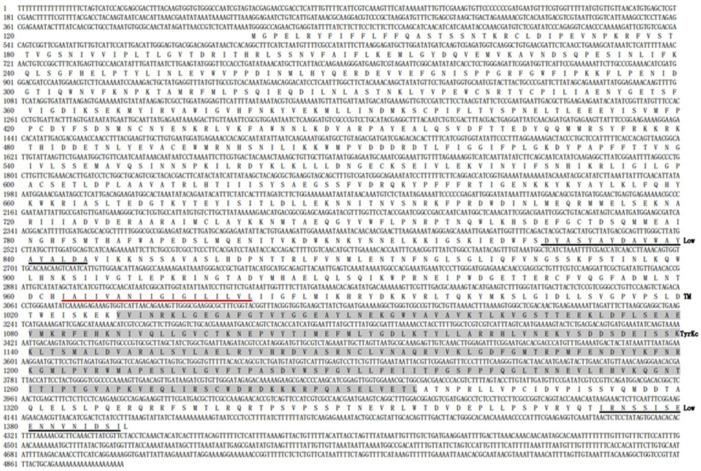
Analysis of the conserved domain of *OnGCY*. Black underlined amino acids: low complexity region (Low); red underlined amino acids, transmembrane reg™ (TM); amino acids with a shaded background; tyrosine kinase, catalytic domain (Tyrkc).

**Figure 2 insects-15-00110-f002:**
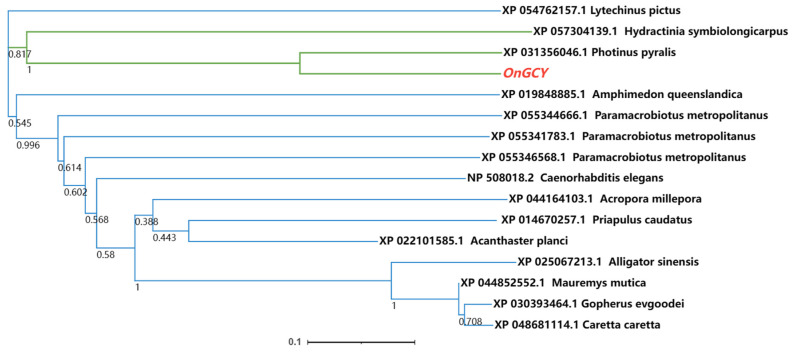
Phylogenetic analysis of *OnGCY* homologs from insect species based on amino acid sequences. Sequences were downloaded from the GenBank protein database. The red font indicates the *GCY* gene of *O. nagaii*. Evolutionary branching lengths are indicated by numbers on nodes of the phylogenetic tree.

**Figure 3 insects-15-00110-f003:**
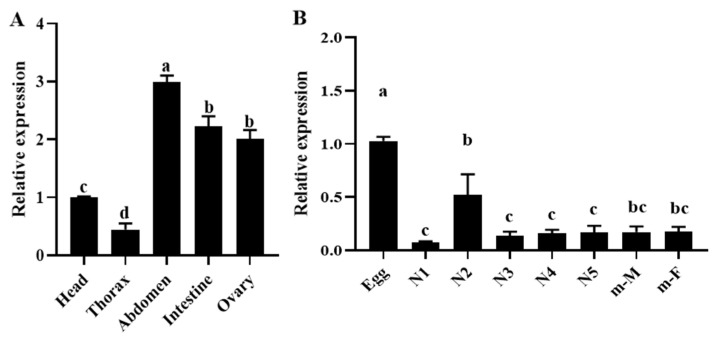
Relative expression levels of *OnGCY* in (**A**) different tissues of 24-h-old female *O. nagaii* adults and (**B**) from egg to adult stages as determined by qPCR. N1, 1st-instar nymph; N2, 2nd-instar nymph; N3, 3rd-instar nymph; N4, 4th-instar nymph; N5, 5th-instar nymph; m-F, mating females; m-M, mating males. Data are mean ± SEM of the mean of three biological replicates and their respective three technical replicates. Different lowercase letters above bars indicate significant differences in the gene expression level between different developmental stages and tissues by one-way ANOVA (*p* < 0.05, Tukey).

**Figure 4 insects-15-00110-f004:**
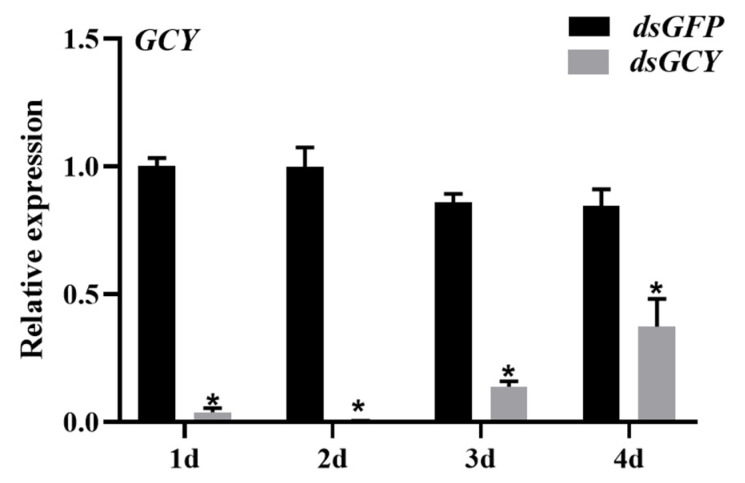
Effects of *OnGCY* silencing at the transcription level. Results are represented as means ± SEM of three independent samples, and samples are normalized to *OnGCY* expression levels. The significance of differences between the treatment group *(dsGCY*) and control group (*dsGFP*) was determined using the Student’s *t*-test for independent samples (* *p* < 0.05).

**Figure 5 insects-15-00110-f005:**
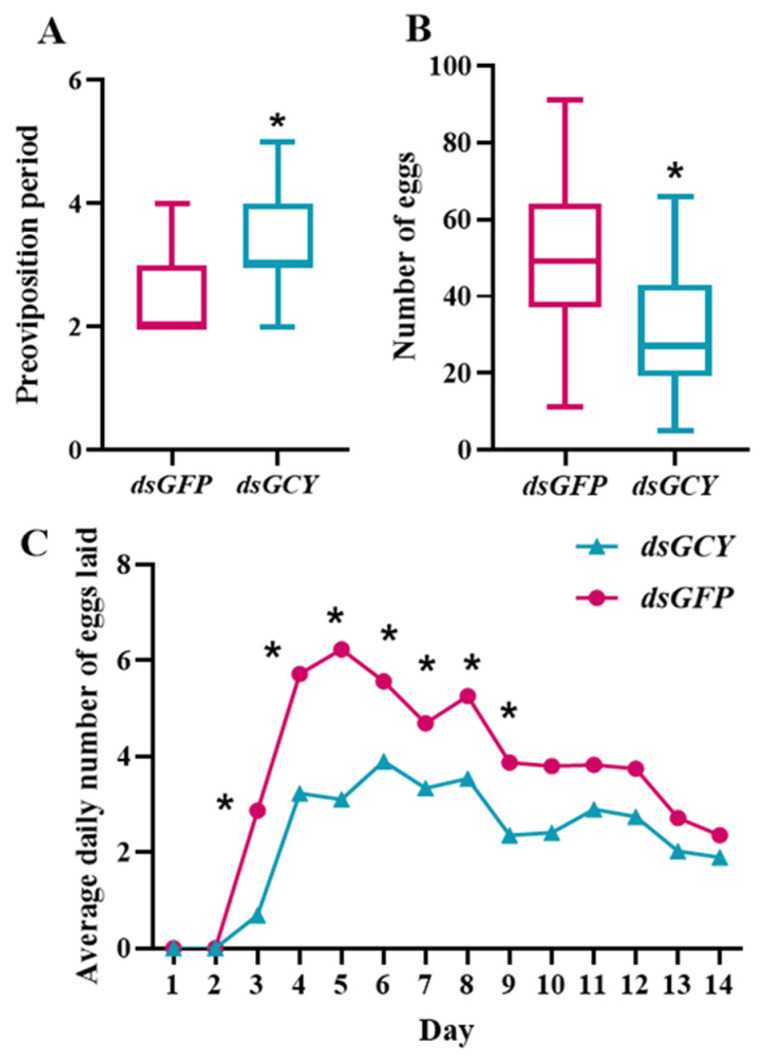
Comparisons of fecundity between the *O. nagaii* treatment group (*dsGCY*) and control group (*dsGFP*). Effects of *OnGCY* silencing on (**A**) preoviposition period, (**B**) number of eggs laid, (**C**) and average daily number of eggs laid. Results are represented as means ± SD of three independent replicates. The significance of differences between the treatment group (*dsGCY*) and the control group (*dsGFP*) was determined in (**C**) using the Student’s *t*-test for independent samples and in (**A**,**B**) using the Mann-Whitney U test (* *p* < 0.05 in both cases).

**Figure 6 insects-15-00110-f006:**
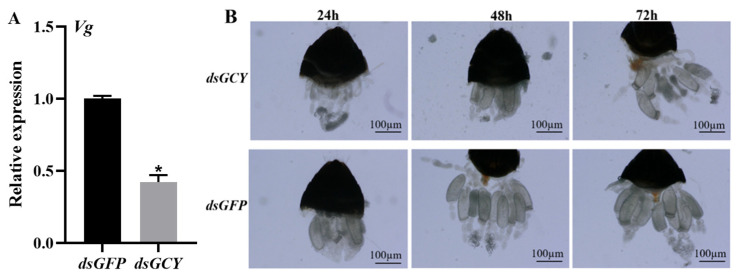
Effect of *OnGCY* on ovarian development of *O. nagaii*. (**A**) Effect of *OnGCY* silencing on *OnVg* expression. (**B**) Effect of *OnGCY* silencing on ovary development after 24, 48, and 72 h. The significance of differences between the treatment group (*dsGCY*) and control group (*dsGFP*) was determined using the Student’s *t*-test for independent samples (* *p* < 0.05).

**Table 1 insects-15-00110-t001:** Primers used for the identification and analysis of *OnGCY*.

Experiment	Primer	Primer Sequence (5′-3′)
RACE	On-GCY-F	ATGGGGCCAGAACTGAGGTA
	On-GCY-R	TTAAAGGATAGAGTCGATGTTAAC
	GCY-3′RACE-f1	AAGCATCGGAGTTGGTGGAA
	GCY-3′RACE-f2-1	TCCGTTCAGATGGACGACAC
	GCY-3′RACE-f2-2	GCTTCGCCAAAGAACACCGT
	GCY-5′RACE-rt-r	ATGAACCATCCGAATCTCCC
	GCY-5′RACE-r1	CGACTTCATCCCTTTCTTGG
	GCY-5′RACE-r2-1	TCCTGTCCGTCACTCCCAAT
	GCY-5′RACE-r2-2	CTTTTTGGGTTGACCTCTGG
dsRNA synthesis	T7-GCY-F	TAATACGACTCATATAGGGGCGCTTAGCTATCTGGCTGA
	T7-GCY-R	TAATACGACTCACTATAGGGGACGATGGAACTGACGGTGT
	T7-GFP-F	TAATACGACTCACTATAGGGCACAAGTTCAGCGTGTCCG
	T7-GFP-R	TAATACGACTCACTATAGGGGTTCACCTTGATGCCGTTC
	GCY-F	GCGCTTAGCTATCTGGCTGA
	GCY-R	GACGATGGAACTGACGGTGT
	GFP-F	CACAAGTTCAGCGTGTCCG
	GFP-R	GTTCACCTTGATGCCGTTC
qPCR	GCY-F	GGTCGCCGTTGCTGTTAAAA
	GCY-R	TGCAGACTCCAAGAAGCTGG
	Vg-F	AGCCTGTTGACTGTCGGAAG
	Vg-R	CGAAGGTCCAACCACTCGAT
	actin-F	CAGAAGGACTCGTACGTCGG
	actin-R	CATGTCGTCCCAGTTGGTGA

Note: Underlined nucleotides indicate DNA sequences transcribed downstream of the T7 promoter. Abbreviations: dsRNA, double-stranded RNA; qPCR, real-time polymerase chain reaction.

## Data Availability

All data are provided within the text.

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
