# Peer review of "Effect of Guanylate Cyclase-22-like on Ovarian Development of Orius nagaii (Hemiptera: Anthocoridae)"

_insects, 2024, doi:10.3390/insects15020110_

Round 1
Reviewer 1 Report
Comments and Suggestions for Authors
The article titled "Effect of guanylate cyclase-22-like on ovarian development of Orius nagaii (Hemiptera: Anthocoridae)" by Hui-Ling Du and colleagues presents a comprehensive study on the effect of the OnGCY gene on the reproduction of the pirate bug Orius nagaii. The research includes detailed methods such as gene cloning, RNA extraction, and bioassays to assess the effects of OnGCY silencing on ovarian development and fertility. The findings demonstrate a significant impact of OnGCY on insect reproduction, providing insights into potential applications in biological control and pest management.
Minor revision,
1. The introduction could benefit from a brief overview of the significance of studying Orius nagaii in biological control.
2, The methods are comprehensive and well-described, providing a good understanding of the experimental setup. The use of RNAi for gene silencing is appropriate, and the statistical analysis seems robust. However, the paper could provide more details on the control measures used to ensure the reliability of the RNAi results.
3, The findings on the role of OnGCY in ovarian development and fertility are compelling. However, the discussion could be strengthened by comparing these findings with similar studies in other insect species to highlight the uniqueness or commonality of the observed effects.
Based on the comprehensive nature of the study, the relevance of the findings, and the potential for significant contributions to the field of biological control, I recommend accepting this paper after minor revisions addressing the points mentioned above. The study is robust, and the minor issues identified do not detract from the overall quality and significance of the research.
Author Response
Dear reviewer,
Thank you so much for your comment on Du et al. 'Effect of guanylate cyclase-22-like on ovarian development of Orius nagaii (Hemiptera: Review of Anthocoridae [Manuscript ID: Insect-2824763]. These comments are of great help to us in improving our paper. We have made changes according to your comments, and the modified part is marked in red font in the text. We hope that this revision will be considered for publication in Insects. Here are our specific responses to the comments:
- The introduction could benefit from a brief overview of the significance of studying Orius nagaii in biological control.
Thank you for your suggestion. Orius nagaii is an important natural enemy insect, which I have described briefly in the background. Green shading indicates. “It can prevent and control aphids and whiteflies on fruits and vegetables, and suck dry prey through pricking mouthparts. Reduce the pollution of chemical pesticides in the field, achieve the effect of green prevention and control, ensure the quality of agricultural products, and have great potential for field application”
- The methods are comprehensive and well-described, providing a good understanding of the experimental setup. The use of RNAi for gene silencing is appropriate, and the statistical analysis seems robust. However, the paper could provide more details on the control measures used to ensure the reliability of the RNAi results.
Thank you for your suggestion. I have made the modification, the modification content is as follows: “Place the O.nagaii in the filter paper so that its abdomen is up near the direction of the glass needle. A NANOJECT II microinjector (Gairdner, Wuhai, China) was then used to inject 50 nL dsGCY (more than 5000 µg/µL) into the middle part of the midfoot coxa and the postfoot coxa of the female adult. Do not pierce the body.”
- The findings on the role of OnGCY in ovarian development and fertility are compelling. However, the discussion could be strengthened by comparing these findings with similar studies in other insect species to highlight the uniqueness or commonality of the observed effects.
Thank you for your suggestion. I have discussed and supplemented them, but there are still too few relevant studies. The research on species is more focused on model organisms, which can only be briefly expounded. The modification is as follows. “In Drosophila, GCY is associated with memory and embryonic cell development [47-49]; In C. elegans, GCY is also involved in body development, but more reports have focused on neural mechanisms [22, 23]. Guanylate cyclase receptors (rGCs) have been found to be involved in the reproduction of basal metazoans in the sperm flagella of coral stones [50]. However, little research has been done on the GCY gene in other species, focusing more on model organisms.”
Reviewer 2 Report
Comments and Suggestions for Authors
Review Report:
The study successfully identified and characterized the gene encoding receptor-type guanylate cyclase-22-like (GCY-22; OnGCY) from the pirate bug Orius nagaii. The investigation utilized rapid amplification of cDNA ends (RACE) to obtain the full-length cDNA, revealing important insights into the molecular structure and biological function of OnGCY. The study also explored the physicochemical properties of OnGCY using ExPASy software and investigated its impact on the ovarian development and fertility of female O. nagaii through RNA interference. The findings shed light on the crucial role of OnGCY in insect fecundity, providing valuable information for agricultural pest control and the large-scale breeding of biological control agents.
Comprehensive Characterization: The study provides a thorough characterization of the OnGCY gene, including its full-length cDNA, molecular properties, and structural features. The use of RACE and bioinformatics tools adds depth to the understanding of this important gene.
Functional Insights: The study goes beyond gene identification by investigating the biological function of OnGCY. The observed impact on ovarian development and fertility, as well as the downstream effects on other GCY and Vg genes, contributes significantly to our understanding of the gene's role in insect physiology.
Practical Implications: The implications of the findings for agricultural pest control and the breeding of biological control agents are highlighted. This practical aspect enhances the relevance and applicability of the study in the field of pest management.
Comments to authors
Areas of Improvement in the Materials and Methods Section:
Clarity in Environmental Conditions: The description of the environmental conditions for rearing O. nagaii lacks specific details. While temperature, relative humidity, and photoperiod are mentioned, providing additional information, such as light intensity and any specific controls for these conditions, would enhance clarity and reproducibility.
Details on Sample Collection: The section describing sample collection could benefit from more precise details. For instance, the rationale behind choosing specific developmental stages and tissues for analysis is not provided. Clarifying the criteria for sample selection and addressing potential variations would improve the reliability of the study.
RNA Extraction Information: Although the RNA extraction process is briefly mentioned, specific details about the RNA extraction protocol used, including any modifications or specific steps, should be provided. This ensures transparency and aids researchers in replicating the study accurately.
Lack of Replication Details: While the term "biological replicates" is mentioned, the section would benefit from a clearer explanation of the number of replicates performed for each experiment. Additionally, details on the selection criteria for individual samples within each replicate would enhance transparency.
RACE Protocol Details: The methods for rapid amplification of cDNA ends (RACE) are outlined, but more specific details, such as primer concentrations, cycling conditions, and gel electrophoresis parameters, would be valuable for researchers attempting to reproduce the experiment.
PCR Amplification Conditions: The PCR conditions for gene cloning and RACE experiments are briefly mentioned, but the cycling parameters and rationale for choosing specific conditions are not fully elaborated. Providing specific details for PCR conditions and justification for parameter choices would enhance methodological clarity.
Gene-Specific Primers: The primer design for gene-specific amplification is discussed, but more information on the rationale behind primer selection, including melting temperatures and sequence considerations, would be beneficial. This information helps validate the specificity of the designed primers.
Incorporating these clarifications and additional details would improve the overall transparency, reproducibility, and reliability of the study's materials and methods section.
Recommendation:
With revisions addressing the identified areas for improvement, I recommend MAJOR REVISION, considering the study's valuable contributions to both basic research and practical applications in pest management.
Author Response
Dear reviewer,
Thank you so much for your comment on Du et al. 'Effect of guanylate cyclase-22-like on ovarian development of Orius nagaii (Hemiptera: Review of Anthocoridae [Manuscript ID: Insect-2824763]. These comments are of great help to us in improving our paper. We have made changes according to your comments, and the modified part is marked in blue font in the text. We hope that this revision will be considered for publication in Insects. Here are our specific responses to the comments:
(1) Clarity in Environmental Conditions: The description of the environmental conditions for rearing O. nagaii lacks specific details. While temperature, relative humidity, and photoperiod are mentioned, providing additional information, such as light intensity and any specific controls for these conditions, would enhance clarity and reproducibility.
Thank you for your suggestion. We use 12W energy-saving lamps. The replacement prey was fresh S. cerealella eggs irradiated with ultraviolet light for 30 min, and kidney beans were used as water source and yield medium.
(2) Details on Sample Collection: The section describing sample collection could benefit from more precise details. For instance, the rationale behind choosing specific developmental stages and tissues for analysis is not provided. Clarifying the criteria for sample selection and addressing potential variations would improve the reliability of the study.
Thank you for your suggestion. This section has been modified and the appropriate content added. Three biological replicates per developmental stage and tissue. The sample size was selected to ensure RNA extraction and reverse transcription
(3) RNA Extraction Information: Although the RNA extraction process is briefly mentioned, specific details about the RNA extraction protocol used, including any modifications or specific steps, should be provided. This ensures transparency and aids researchers in replicating the study accurately.
Thank you for your suggestion. The extraction of RNA followed the experimental procedure completely. Detail the first step of sample extraction and the last step of RNA collection. The intermediate steps follow the instructions exactly, with no changes.
(4) Lack of Replication Details: While the term "biological replicates" is mentioned, the section would benefit from a clearer explanation of the number of replicates performed for each experiment. Additionally, details on the selection criteria for individual samples within each replicate would enhance transparency.
Thank you for your suggestion. I have revised the original text in detail. Three biological replicates were performed for each test sample. Fluorescence quantitative tests were performed four times for each sample and three times for biological repetition
(5) RACE Protocol Details: The methods for rapid amplification of cDNA ends (RACE) are outlined, but more specific details, such as primer concentrations, cycling conditions, and gel electrophoresis parameters, would be valuable for researchers attempting to reproduce the experiment.
Thank you for your suggestion. Primer concentration follows instructions, picture below.
All trials were 30 cycles. The electrophoresis parameters were 180V, 20min.
(6) PCR Amplification Conditions: The PCR conditions for gene cloning and RACE experiments are briefly mentioned, but the cycling parameters and rationale for choosing specific conditions are not fully elaborated. Providing specific details for PCR conditions and justification for parameter choices would enhance methodological clarity.
Thank you for your suggestion. The test was carried out in full accordance with the instructions and only the annealing temperature was changed. We use the software primer5 to design the primer, the length of the primer is usually between 21-25. RACE round 1 and round 2 annealing temperatures are fine-tuned between 60 and 65 degrees
Gene-Specific Primers: The primer design for gene-specific amplification is discussed, but more information on the rationale behind primer selection, including melting temperatures and sequence considerations, would be beneficial. This information helps validate the specificity of the designed primers.
Thank you for your suggestion. The unity of the strip was determined by electrophoretic glue running, and the peak was determined by the dissolution curve.

Round 2
Reviewer 2 Report
Comments and Suggestions for Authors
The authors have addressed all of my questions. I recommend to accept the MS in present form.